# Quantitative Framework for Bench-to-Bedside Cancer Research

**DOI:** 10.3390/cancers14215254

**Published:** 2022-10-26

**Authors:** Aubhishek Zaman, Trever G. Bivona

**Affiliations:** 1Department of Medicine, University of California, San Francisco, CA 94158, USA; 2UCSF Helen Diller Comprehensive Cancer Center, University of California, San Francisco, CA 94158, USA; 3Chan-Zuckerberg Biohub, San Francisco, CA 94158, USA

**Keywords:** quantitative biology, chemical biology, bench-to-bedside, OMICS, IC50, high throughput screen (HTS)

## Abstract

**Simple Summary:**

Technological advancements and emerging high throughput molecular data have transformed biology into a more quantitative and multidisciplinary discipline. This has accelerated the translation of laboratory based findings into applied and clinically relevant applications and therapeutics. A shared practice for quantifying and statistical rank-ordering the effects of such translational applications and for understanding their underlying mode-of-action is now critical. In this manuscript, we discuss some of the major types of quantitative translational research and the best practices. We propose that adherence to these guidelines will improve assay design and reduce missteps in translational biomarker and therapeutics clinical application and adoption.

**Abstract:**

Bioscience is an interdisciplinary venture. Driven by a quantum shift in the volume of high throughput data and in ready availability of data-intensive technologies, mathematical and quantitative approaches have become increasingly common in bioscience. For instance, a recent shift towards a quantitative description of cells and phenotypes, which is supplanting conventional qualitative descriptions, has generated immense promise and opportunities in the field of bench-to-bedside cancer OMICS, chemical biology and pharmacology. Nevertheless, like any burgeoning field, there remains a lack of shared and standardized framework for quantitative cancer research. Here, in the context of cancer, we present a basic framework and guidelines for bench-to-bedside quantitative research and therapy. We outline some of the basic concepts and their parallel use cases for chemical–protein interactions. Along with several recommendations for assay setup and conditions, we also catalog applications of these quantitative techniques in some of the most widespread discovery pipeline and analytical methods in the field. We believe adherence to these guidelines will improve experimental design, reduce variabilities and standardize quantitative datasets.

## 1. Introduction

“What is life?”, once asked quantum physicist and Nobel laureate Erwin Schrödinger, when he prophesized that behind the mystical nature of life there must remain a quantifiable mathematical elegance [1]. Not long after, James Watson and Francis Crick presented the first mathematical model for the ‘molecule of life’-DNA [2]. Ever since then, from a systems biology point of view, the quest to generate mathematical models to quantify biological processes as reactions, to quantify stimuli and response as input and output has emerged. Over the years, due to a remarkable expansion in high throughput data acquisition and in our ability to analyze, biological research has become vastly more quantitative. In vitro research coupled with powerful statistical analysis have successfully recapitulated patient and in vivo biology indicating the power of quantitative biology in the field of biochemistry, molecular and cellular biology and cancer research and treatment. This has also caused a paradigm shift towards massively automated and computation-heavy annotation and analysis in many disease contexts such as cancer biology [3]. For example, quantitative chemical biology research has transformed our understanding of tumor genomics, prediction of novel candidate therapies against cancer and improved personalized targeted therapy for cancer patients. In last decade alone, worldwide chemical biology efforts have resulted in an avalanche of clinical trials and FDA-approved therapies against many different types of tumors, subsequently increasing patient longevity and decreasing therapy toxicity [4].

The broader goal of quantitative chemical biology research is to model the pharmacokinetics and pharmacodynamics of chemicals in diseased patients, but in a much more conducive and tractable in vitro system initially. Modern chemical biology research takes advantage of our ability to screen unbiasedly through a wide array of compounds in systems such as in patient-derived cell lines, patient derived xenografts (PDXs) or purified protein-ligand binding assays [5]. Furthermore, modeling drug response by high throughput screening (HTS) is followed by identifying candidate biomarkers, signaling pathways and molecular targets. Chemical biology research also focuses on improving the response of the drugs through modifications of chemical structure followed by quantitative structure activity relationship (QSAR) studies or through drug synergy assays. Quantitative chemical biology is making a reality what once was science fiction [6]. Yet, there are gaps that must be addressed to better translate quantitative cancer research into clinical implementation. Here, in this review, we outline individual aspects of chemical biology research and their quantitative frameworks. We also summarize progress in this area of cancer research, highlight key gaps, and propose concrete steps forward.

## 2. Modeling Drug Dose Response

The goal of quantitative biology is to quantify biological processes and chemical biology to discern the effect of chemicals on biological systems. Proteins are the functional molecules of life, important for carrying out most biological reactions and in return for steering biological processes. Hence, a major focus of chemical systems biology is to model the response of different doses and kinetics of chemical perturbagens on enzymes.

For example, enzyme inhibitors are often pharmacological agents that competitively and reversibly inhibits substrate binding and enzyme activity [7]. The kinetic behavior for many enzymes can be explained with a Michaelis-Menten (MM) model for enzyme-ligand or enzyme-substrate binding and catalysis:$$E+S\begin{array}{c} k_{1}\\ ⇌\\ k_{-1}\; \end{array}\;\mathrm{ES}\;\overset{k_{2}}{\to }\;E+P$$
where E stands for enzyme, S stands for substrate and P stands for product. ES is an enzyme-substrate complex that is formed prior to the catalysis. Formation of ES requires only binding and hence is reversible, indicated by equivalent rate constant k_1_ and k_−1_ for forward and reverse rate for the event. On the other hand, the overall rate-limiting and irreversible step in the reaction is the breakdown of the ES complex to yield product, which can proceed with rate constant k_2_.

Reaction velocity from this reaction can be described as a function of substrate concentration using the following formula, which is typically referred to as the Michaelis-Menten (MM) equation:v = ([S]Vmax)/([S] + Km)
where, v = rate of reaction during initial velocity condition; Vmax = maximal reaction rate; S = substrate concentration; Km = Michaelis-Menten constant. Interestingly, since the value of Km at 0.5 Vmax condition equals to [S], Km is often termed as the substrate concentration at half maximal velocity (Figure 1a).

A MM equation is different from an enzyme substrate reaction progress curve that often describes a kinetic equilibrium. It is important to note that the MM model plot in Figure 1a does not describe activity of the enzyme under a continuous time variable; instead, velocity, v is calculated separately for corresponding individual substrate values (Figure 1b) [8]. Hence, many different measurements of v are calculated each under initial velocity conditions with varying substrate concentrations at or below the Km value (Figure 1a). Measurement at the initial velocity conditions ensures that the equation is insensitive to the effect of velocity variation during reaction progression. However, the equation predicts saturation of reaction rate at Vmax and an initial logarithmic increase in velocity as a function of substrate concentration (Figure 1a) [7].

Formation of product, post-enzyme-substrate mixture is the quantitative estimate of reaction rate and when tracked and plotted over a period indicates the reaction progress rate (Figure 1b). However, initial velocity of the enzymatic reaction is separate from the velocity of reaction during reaction progression. Initial velocity represents the reaction rate when less than 10% of the substrate has been depleted or less than 10% of the product has formed. Under these conditions, it can be safely assumed that the substrate concentration does not significantly change and does not reach saturation limit for enzymatic activity. Furthermore, it can also be assumed that, in such conditions, the contribution of reverse reaction is minimal [8].

The MM equation is a first order reaction which results in a linear increase in v until the reaction saturates due to maximal occupancy of enzyme with substrate. This is a classic example of 1:1 substrate enzyme interaction consistent with the ‘lock and key’ model. However, for enzymes with multiple substrates, cooperativity amongst the substrates for enzyme binding and requirement of maximal enzyme site occupancy for enzyme activity results in an initial lag time for reaction initial velocity followed by an exponential increase in enzyme activity and subsequent saturation. In this case, the reaction follows a multi-order non-linear and sigmoidal reaction kinetics, often consistent with a so-called ‘induced fit’ model, determined by what is called a hill coefficient. The higher the hill coefficient the sharper the inflexion of the hill curve (Figure 2) [9].

## 3. Determination of IC50 for Inhibitors

The measurement of enzyme activity in the linear initial velocity condition is often time not possible due to multiple reasons, lack of a measurable assay read out being one of them. In these cases, the enzyme or a linked biomarker response is measured at a fixed time point as a function of varying substrate concentration. This concentration-response plot is similar to the Michaelis-Menten plot except that response is not calculated at initial velocity condition (Figure 3a). Furthermore, for enzyme inhibitors the curve simply follows an upside-down version of the concentration response plot for an enzyme substrate (Figure 3b) [10]. Effects of an inhibitor on enzymatic reaction can often be determined by plotting a dose/concentration response plot where *x*-axis represents varying doses of the inhibitor and *y*-axis represents corresponding reaction rate estimates. Notably, in this setting, the enzyme and substrate concentration are kept constant. The dose response plots are widely applied in pharmacology and chemical biology. For example, it is often the first assay to be performed to identify a candidate list of active/lead compounds by screening a library of known or unknown ones. It is also applied for determination of ideal dose range and therapeutic window as well as for structure activity relationship (SAR) assay for chemical/functional group variation of a desired compound’s backbone for activity enhancements. C The 4-parameter logistic nonlinear regression model (4PL) for data fitting, that describes the sigmoid-shaped response pattern, is an example of this type of model (Figure 3a,b) [10]. For example, in Figure 3a light blue curve represents a concentration response plot for an enzyme inhibitor.

Analogous to MM constant (Km), substrate concentration required to result in 50% activity is called EC50 (effective concentration to reach 50% activity). On the contrary, the concentration of compound that results in 50% inhibition of maximal activity is termed the IC50 (inhibitor concentration yielding 50% inhibition) [11]. In this review, we discuss the IC50 calculation often deployed in cancer biology, where instead of inhibitor binding to the target of interest (target-based), biologists measure the inhibitor response on cellular viability (phenotype/cell based).

Some criteria for successful concentration-response curves are listed in the discussion below.

Well defined top and bottom plateau values need to be established. To do so, it is important to use sufficient range of inhibitor concentrations. These parameters are critical for the mathematical models used to fit the dataA minimum of 8–10 inhibitor concentration data points for an accurate IC50 determination should be usedConcentration ranges for the inhibitors should be spaced equallyThe concentration data point counts and the range should be chosen so that half the data points on the IC50 curve are above the IC50 value and half are below the IC50 value. This is difficult for IC50 measurements for compounds for which there exist no prior knowledge. In this case, the inhibitors should be tested for response using a broader range of doses followed by final IC50 estimation using narrower range of dosesEnzyme concentration should always be kept constant and the lower limit for determining an IC50 is half of the enzyme concentrationWell readable and quantifiable screening strategies for measuring the response should be employed. The quantification should be benchmarked under different experimental conditions. For example, cellular viability can be measured by viable cell adenosine triphosphate (ATP) level using the reagent cell titer glo (CTG)At least three replicates for each data point should be collected. For cellular viabilities these replicates need to be biological replicatesCriteria for reporting IC50′s are the maximum % inhibition should be greater than 50%; top and bottom values should be within 15% of theory; the 95% confidence limits for the IC50 should be within a 2–5-fold range. Relative and absolute IC50 and EC50 is described in Figure 3b.

Depending on necessity, a wide array of reagents can be used as a replacement of CTG for cell viability measurements. For example, dye coating and dye exclusion -based experimental setup requires use of crystal violet, trypan blue, eosin, congo red and erythrosine B staining. For non-ATP and nicotinamide adenine dinucleotide phosphate (NADPH) -based colorimetric assay MTT, MTS, XTT, WST staining reagents are used (PMID: 9869118, 28470513) [12,13]. For cellular protein and enzyme level as a proxy for viability LDH and SRB assays are widely accepted. On the other hand, alamar blue and CFDA-AM are two commonly used fluorometric cell viability assay (PMID: 28470513) [13]. Additionally, immunofluorescence and flow cytometry-based assays (e.g., Brdu, annexin 5) are also commonly used for the purpose of determining viable cells (PMID: 28573164) [14].

The concentration-response curve response does not plateau at the baseline (e.g., 0%) or does not saturate at the highest point (e.g., 100%). This may happen due to inter sample heterogeneity (e.g., bimodal response samples) or due to technical issues. In this case, the theoretical IC50 values is different from the test IC50 values calculated, giving rise to inaccurate IC50 estimates. In those cases, area under curve (AUC) calculation can offer a more accurate estimate of the response [15].

## 4. HTS Using Pharmaco-Chemical Library

A holy grail in oncology is so-called ‘magic bullet’ therapies that perturb only diseased cells/proteins but leave normal healthy cells/proteins untouched. Over the course of time, our understanding of proteins, as the functional molecules of cells, has immensely improved and has resulted in interest to target them in pathophysiological conditions [16]. For example, we have undertaken technologically sophisticated high throughput screening for pharmacological compounds that perturb or ameliorate the activity of a protein molecule and thereby correct a disease phenotype.

In such a drug screening experiment, the efficacy of several pharmacological agents are evaluated either against a disease phenotype or against an enzyme activity. The former is called a phenotypic screen whereas the latter is known as the target-based screen. The goal of these screening approaches is to identify, from a wide an array of initial compound list, a smaller and tractable number of candidate compounds (often called ‘leads’) [17].

To find novel therapeutics against a disease, phenotypic screening, where a myriad compounds are tested for reduction of a disease phenotype, is most commonly employed. Phenotypic screening is unbiased and agnostic about the mechanism of action (MOA) or the molecular target for the tested agents. Therefore, subsequent analysis for target deconvolution is required for comprehensive understanding of the effect of the compound. In this regard, a more focused version of screening called target-based screening can be applied. wherein this case, pharmacological agents are tested against a single or handful of molecular targets; targets that have already been identified as a causal mechanism of disease pathology. Target-based screenings are usually less time consuming, but difficult to design. Moreover, target-based screening approaches are a non-starter for diseases for which a knowledge deficiency exist (Table 1) [18]. Both approaches have their pros and cons. Below is a comparison of phenotypic screenings and target-based screening (Table 1):

For HTS, the concept of combinatorial chemistry was developed in the mid 1980′s, with Geysen’s multi-in technology where hundreds of thousands of peptides were synthesized on solid support in parallel [19,20]. Subsequently, one-bead one- compound (OBOC) combinatorial peptide libraries and solution-phase mixtures of combinatorial peptide libraries and phage display libraries were introduced. However, it was not until mid-90 s, when the first example of a small-molecule combinatorial library was reported [21].

Combinatorial chemistry has been used for both drug lead discovery and optimization. The highly focused parallel synthesis of small-molecule libraries (hundreds to thousands of compounds), when developed in conjunction with computational chemistry, are particularly useful for optimization of drug leads [10].

Recommendations and challenges for IC50 calculations in HTS:

Doubling time: IC50 is best calculated in an isogenic setting, where response of the cells to a particular perturbation is best compared with that of a response without the perturbation. However, in absence of an isogenic system, classifying IC50 spectrum of many different cell lines into high and low, leads to a possibility that the difference in IC50 is due to doubling time differences (Figure 3a). For example, cells with higher metabolic activity and doubling time, are prone to up taking the compound faster and hence will be killed faster, resulting in an IC50 smaller than cells that grow slower. Hence, regression analysis of IC50 and doubling time is required to rule out this phenomenon where, ideally, no significant correlation between the doubling time and IC50 is preferred.

Number of cells or seeding confluency: To balance for the doubling time often cells are seeded in a manner so that by the time they are ready for measuring the effect of the drug they are of around 90% confluency. Quite intuitively, it has been observed that, higher confluency of seeding requires a higher dose of compound to kill 100% of the cells. The effect is often described as a drug sync/sponge effect. Hence, the initial seeding densities of the cells required, needs to be accurately estimated by empirical trials [10].

Edge effect: Screening platforms often use small multi-well formats (e.g., 384 and 96-wells). It has been reported that the wells in the plate that are situated at the edge of the plates are exposed to external stimulus such as temperature un-uniformly than that of the wells in the middle. Hence, edge wells are usually exempted from using during the IC50 calculation in these plates.

Vehicle effect: Each perturbation must be compared with a vehicle treatment cohort. Often the Vehicle treatments can result in some response alone (Figure 3b). At long as the response is below 10%, the response is considered acceptable. The vehicle can also result in some confounding cellular effect above a certain dose; hence it is very important to keep the vehicle dose within the limits of acceptable range. The lead compounds being tested should have the same amount of vehicle in volume for comparative analysis. If compounds involved in a screening assay were dissolved in various vehicles, the screen must consist of many different ‘vehicles alone’ controls for comparative analysis as well.

## 5. Biomarker Prediction

One of the major limitations of phenotypic screens is the lack of understanding of any molecular targets for the drug itself. Hence, target discovery from phenotypic screen has been a major challenge in the field of chemical biology. Since the advent of high throughput genomics technologies, many computational approaches have been undertaken to correlate drug phenotype response to cellular genomic, epigenetic, transcriptomic, proteomic and metabolic features [22,23,24].

National Cancer Institute (NCI) initiated a Drug sensitivity Dialogue for Reverse Engineering Assessment and Methods (DREAM7; Available online: http://dreamchallenges.org/ (accessed on 12 August 2022) project to gather momentum and bolster enthusiasm for this very important challenge of predicting biomarkers from drug sensitivity and vice versa. The Challenges is one of the first of kind- a community-based collaborative competition oriented towards crowdsourcing solution and open-data sharing [22,23]. The DREAM7 Challenge also benchmarks many drug sensitivity prediction methods. For example, kernel-based prediction methods, which depend on machine learning algorithm for pattern matching (e.g., support vector machine or SVM), differ from feature-based methods, which depends on a feature map generated by training dataset, in terms of utilization of the user-defined feature map [22,23,25,26].

In NCI-DREAM7 Challenge, for training datasets a multi-OMICS (e.g., copy number variation, DNA methylation, point mutations, transcriptional and protein level estimates) approach was pursued. Interestingly, the predictive models that used multi-omics profiles outperformed a single-OMICS prediction model, which suggests genomic, epigenomic, and proteomic profiles provide complementary signal for drug response prediction [24,27,28]. Importantly, prediction algorithms validated previous biological knowledge for breast cancer and provided insight into non-linear feature relationships during modeling [22,23].

One useful approach is a regularized regression model known as elastic net [29]. One of the major problems of these biological datasets is the asymmetry of the matrices. The columns of the matrices containing various treatments (<200) were much too small in number than the rows of the matrices that contain genomic features (>5000), which poses a computational challenge for regression model often known as ‘*p* >> *n* ratio problem’ [24,25,30]. Ordinary regression models due to this asymmetric in matrices generates overfitting solutions resulting in false positive/type1 errors. To solve this, the elastic net generates sparser biomarkers based on a regularized regression model where the equation balances between lasso and ridge regressions. Furthermore, the resulting solutions can be represented as a heatmap (Figure 4a) [15,30,31,32].

The best-performing algorithm was based on the Bayesian efficient multiple kernel learning (BEMKL) model. BKMEL uses a kernelized regression model that makes use of both multi-task and multi-view learning algorithms [23,26]. In Multiple kernel learning (MKL) algorithm, pairwise similarities of cell line OMICS profile constitute an initial kernel and are subsequently combined into a compound kernel. In multi-task learning (MTL), on the other hand, the model is trained simultaneously for all the drugs and thus differ from the stepwise kernel generation strategy employed in MKL. BKMEL introduces hyper-parameters and an error term/bias to account for poor intersection of multi-OMICS datasets [22,23].

## 6. IC50 Measurements in Isogenic Settings

Biomarker prediction depend on a correlation between features and drug response. However, these methods do not essentially establish causality. To address this, incorporating tumor associated alterations in an isogenic system is increasingly being pursued for comprehensive chemogenomic analysis [33]. In this setting, a patient derived cell line or organoid of interest is genetically subjected to very specific genetic modifications and subsequently drug responses are measured across the board to determine the effect of the genetic alterations. The resulting viabilities can be represented as a heatmap (Figure 4b). On many occasions, these have contributed crucial understanding of oncogenic addiction, specificity of crosstalk between pathways and genetic interactions in cancer. Recently, the ease of activating or perturbing genetic alterations using CRISPR based technologies have paved the way for new opportunities for high throughput chemogenomic interaction analyses [34,35,36,37,38,39,40].

## 7. Signaling Pathway Analysis and Target Discovery

In biological response versus feature correlation analyses, instead of enrichment of a single biomarkers, enrichment of a list of functionally related group of genes is more informative. As a result, increasingly the classical gene-based approaches that ignore the modular nature of most human traits is being replaced with a more functionally holistic pathway enrichment approach. In this regard, statistically computing overlapping between experimental OMICS datasets (such as exome, methylome, RNAseq, quantitative proteomic, metabolomic, etc.) and curated pathway databases (e.g., GO, ENCODE, KEGG, REACTOME, etc.) have become routine [41,42,43,44,45]. The computational analyses depend on either hypergeometric tests (ENRICHR) or a signal to noise based (S2N) analysis (BROAD institute GSEA) [45,46,47].

In pathway analysis, a set of candidate/query genes are compared against a library of curated ‘gene sets’ each of which includes genes that are bundled due to their participation in a signaling pathway or biological function). The candidate/query gene sets are usually composed of genes that are differentially upregulated or downregulated in an OMICS dataset such as gene expression, proteomics, etc. (Figure 5). The prototypical enrichment or overrepresentation (ORA) analysis is usually performed via comparing the test gene set with that of the curated gene set using hypergeometric test, where null hypothesis represents a baseline or random-chance representation probability [48,49,50,51]. Overlapping is considered significant if the hypergeometric test produces a significant *p* value (Figure 5). Hypergeometric test can be done in presence or absence of weighted or ranked gene sets as well. However, this method does not consider the topology of a signaling network. In biological pathways, genes/proteins tend to perform in a network where each gene/protein can be thought to be as a ‘node’ (drawn as circles) and their regulation between one another is signified by ‘edges’ (drawn as lines) (Figure 5). From this network point of view, minor variation in gene/protein neighborhood and directionality of reaction contributes to a vastly different biological function. Hence, while considering overrepresentation, curated databases that incorporate directionality and neighborhood information (such as Reactome, KEGG, WikiPathways, etc.) produces better signaling pathway analysis (Figure 5) [42,43,52]. More recently, even more granular context specific sub-circuitry and subnetwork based over representation analysis have become increasingly useful and is called mechanistic pathway activity (MPA) based pathway analysis. For this analysis, the curated database not only has the topology information it also has positivity and negativity information for the nodes in play (Figure 5) [53,54,55,56].

## 8. Form Pathway to Target Discovery

Pathway analysis, based on correlation, generates a single or a few candidates signaling pathways as target of the drug. However, it neither establish causality nor pinpoint a single gene/protein as a target. Hence, for target discovery the goal is to home in on a single protein–protein interaction (PPI) from protein interaction signaling networks; a single enzyme target for cascade of enzymatic reaction; a single gene target for a gene regulatory network [34,57,58]. Causal relationships of a perturbagen often needs to be established by genetic manipulation of the candidate genes/proteins one at a time [59]. Recent technological advances, including genomics, proteomics, small interfering RNA CRISPR, and mouse knockout models have allowed us to measure the effect of shortlisted candidate pathway regulators on the cellular phenotype, which allows us to identify a targetable protein [18,59].

Computational biology and structural biology have been instrumental in deciphering drug-protein and protein–protein interactions in absence and presence of the perturbagens. If a single protein is thought as a hub, then the protein that interacts with it and forms a PPI network constitute the interactome for the protein of interest [60,61].

Structural biology is crucial for PPI research. X-ray crystallography, protein-based nuclear magnetic resonance (NMR) spectroscopy has made it possible to generate and study 3D structure and active site pockets of protein molecules. Computer simulated docking of perturbagens, often known as in silico docking, often faithfully recapitulates the biological ligand/inhibitor binding pocket on the protein [58,61]. However, 3D structures generated through these methods often are static and fails to recapitulate the dynamic nature of the protein-ligand interaction. Furthermore, crystallization of the protein itself requires many modifications of the protein molecule such as truncation and/or mutations [58]. Although far from perfect, in the last few years, significant improvement in dynamic structural simulations such as monte carlo simulations have raised remarkable promise for in silico simulations for protein-ligand/inhibitor interaction [62].

## 9. Quantitative Structure Activity Relationship (QSAR) and Physicochemical Properties of Drugs

Quantitative Structure-Activity Relationship modeling is one of the major computational tools employed in medicinal chemistry [63,64]. In QSAR analysis a structural element (called a molecular descriptor) of the lead chemical compound is modified and the response in activity is measured [65]. The goal, this way, is to generate an array of activity response and curate and finetune the best response. Modification of molecular descriptor of the compound can be based on its chemical 2D structure as well as 3D topography [63,66].

One of the major challenges with QSAR equations is that of faithfully predicting the effect of multiple modifications at once. For example, two colinear molecular descriptors independently may result in improvement in QSAR response; however, when introduced together may result in antagonistic response. This indicates the importance of experimental validation of QSAR response to avoid such confounder effect.

Moreover, applications of the concept of drug-likeness, which compares physical properties of candidate pharmacological chemicals (such as lipophilicity) with that of other validated compounds and approved drugs to predict pharmacodynamics and pharmacokinetics of the candidate drug [66]. These in silico predictions help both the final in vivo preclinical and clinical validation experiments, by helping decide the range of doses and time-period to be tested.

## 10. Drug Synergy

For any complex disorders, rational design of multi-targeted drug combinations is a promising strategy not only to improve individual drug potency and efficacy, but also to tackle resistance to individual drugs. A drug combination is usually classified as synergistic or antagonistic, depending on the deviation of the observed combination response from the expected effect calculated based on a reference model of non-interaction [67]. There are many metrics for drug combination measurements. Combination effect measurements can vary due to the experimental design. For example, before the advent of high throughput platforms single dose combination therapy was widespread [15,40,68]. However, the field has moved towards a much more sophisticated and comprehensive methods of combining the doses of the drugs (Figure 6) [68,69,70,71].

Often, synergy is calculated by generating comprehensive dose response under a varying level of Drug 1 and Drug 2 (Figure 6a) [72]. These generate a symmetric matrix and often called checkerboard dose combinations (Figure 6b). However, due to effort intensive nature of this setup, synergy is often calculated by measuring dose response curve of cells under Drug 2, in the presence of a few doses of a test compound, Drug 1 [68,72,73]. The graph looks like few dose–response curves in the same plot and they superimpose in top of each other in absence of synergy or antagonism and shift left or right in presence of synergy and antagonism, respectively, (often called a multiple-ray plot). Synergy can also be measured using just a single dose of Drug 1 and Drug 2. However, quantification of synergy under those condition is difficult [72,73]. For any two compounds Drug 1 and Drug 2, following is a summary of ways for quantifying the synergy.

Statistical independence/Bliss: This quantification is applicable for synergy calculation even using a single dose. However, Synergy calculation using a few doses, or a single dose is also possible. This can be calculated by calculating the probability/percentage of killing under each individual drug treatment-
P_a+b_ = 1 − (P_a_ × P_b_)
where P_a_ = probability of killing cells by drug 1; P_b_ = probability of killing cells by drug 1 and P_a+b_ = probability of killing cells together.

The other quantification methods for synergy are quantified by measuring and adding maximum response by each drug alone and then measuring the effect of the combined dose. If combined effect of maximum dose is more than that of the additive effect, the interaction is called synergistic.

Gaddum pharmacological interaction: In presence of the Drug B; the dose response curve of Drug 1 shifts on the left and the new IC50 value is α, lower than that of the IC50 in absence of Drug 2. Gaddum pharmacological interaction measures difference in this IC50 as a measure of synergy (Figure 6a) [74].

Isobologram analysis: In checkerboard comprehensive synergy analysis, a response-surface plot generated as such allows generation of isobologram graphs (such as contour plots in geography) by connecting identical level of toxicities in the different drug combinations [75,76]. For example, if Drug 1 alone causes 20% toxicity and Drug 2 causes 20%; Drug 1 and 2 together causes 40% toxicity then the isobologram would go through a rectangle line as all the 20% toxicity values would fall in that isobologram line. However, this visualization does not aptly explain the additive effect of two drugs (Figure 6b,c).

Lowe additive model analysis: In comprehensive analysis, Drug 1 will interact with Drug 1 as in an additive manner [77,78]. From this expectation, If Drug 1 and Drug 2 behaves as though their effect is additive the isobologram would go through a diagonal line. Similarly, if the combination results in better response than each compound alone then the isobologram would go through a concave line and similarly for antagonistic interaction, a convex line.

The Chou Talay combination index from this can be calculated using the formula for this is (Figure 6c):$$\mathrm{Combination}\;\mathrm{index}\;\left(\mathrm{CI}\right)=\frac{a}{IC50\_A}+\frac{b}{IC50\_B}$$

CI > 1 means antagonism; CI < 1 means synergistic interaction and CI = 0 means additive interaction (Figure 6c) [79]

## 11. Case Study

In this section, we now describe an experimental example that illustrates the concepts discussed above. In this example, an investigator found a cellular receptor tyrosine kinase called Fibroblast Growth Factor Receptor 1 (FGFR1) and its downstream pathway to be upregulated in a subset of non-small cell lung cancer (NSCLC) samples (Figure 5). Additionally, the investigator found that genetic suppression of FGFR1 reduces the viability of NSCLC cells by performing crystal violet and cell titer glo (CTG)-based viability assays (described in Section 3). Hence, the investigator concluded that this particular subset of NSCLCs required FGFR1 for cell proliferation.

Establishment of this causal relationship between the FGFR1 pathway and cell viability motivated the investigator to next test existing chemical perturbagens of the FGFR1 pathway, such as Infigratinib, using the CTG assay (Figure 3). Unfortunately, the investigator found that although the effect is specific, the response to the FGFR inhibitor has a narrow therapeutic window indicated by some toxicity even in normal cells and also requires a high dose for therapeutic effects. Hence, the investigator decided that one next step would be to design an improved inhibitor for this pathway of interest.

The next goal was to test a library of chemical compounds which contain both structurally diverse as well as structurally similar compounds with slight functional group variation compared to Infigratinib. The library was tested in a panel of ~100 NSCLC patient-derived cell lines with 3 replicates for each data point and 12 doses of 3 times serial dilution from the highest dose. The highest dose used for any compound is empirically determined (e.g., 33 µM for Infigratinib). The compound response patterns were also clustered in an unbiased manner. In an ideal case, this cluster should reflect biologically relevant information. For example, small molecules with shared MOA should cluster together. As expected, AZD4547, another FGFR inhibitor was in the same functional clade as Infigratinib. A candidate list of 10 compounds was rank ordered from this experiment and further validated in vitro. These 10 compounds include candidate leads from both the same (such as other FGFR inhibitors) and different functional clades (such as AKT inhibitor MK-2206) as Infigratinib.

Leads that parsed into a different functional clade (and hence employ different MOAs), and yet worked potently against FGFR1 upregulated NSCLC provided novel insights. For example, these small molecules might exhibit on-target blockade of the FGFR pathway up or downstream of the FGFR receptor tyrosine kinase (e.g., AKT inhibitor MK-2206). Additionally, some small molecules could also function in a synthetic lethal-like manner, where they exhibit potent responses against FGFR1 upregulated NSCLC by acting indirectly against additional pathways required for NSCLC tumor survival in this molecular context (e.g., PARP inhibitors).

Although the compound screen was informative in identifying novel leads, no single agent perturbation was sufficiently potent to elicit a complete response (near 100% cell death). Hence, the investigator next applied these 10 lead compounds in dual combinations (10 × 10 = 100 unique combinations) and then measured the viability response again in a high throughput manner. The compound combination effect was then mathematically classified as synergistic, antagonistic, or additive as explained above (Figure 6). The investigator identified potent synergy between FGFR and Mitogen-activated protein kinase (MAPK) pathway inhibitors (e.g., Trametinib). Combined application of these small molecules exhibited a more complete effect against FGFR1 upregulated NSCLC cells.

Interestingly, this compound screen could also be carried out in an unbiased manner in the absence of the pathway information mentioned above. Given the availability of molecular feature dataset for each individual cell lines (such as genomic, transcriptomic, proteomic and/or other), mathematical models can then be applied to uncover associations between cell viability response and genomic feature variables (Figure 4). In this way, it is also possible to identify therapeutic and MOA hypotheses in an unbiased manner even in the absence of any candidate pathway a priori.

We use this example to demonstrate scenarios where these tools can be directly applied. There are many possible combinations of ways these techniques can further be used in the field.

## 12. Bench-to-Bedside Translation

Preclinical research, despite certain limitations, is a powerful approach to predict in vivo clinical responses [5,80]. However, translating observations identified in preclincial systems into actionable therapies in vivo involves a range of additional validations. For example, pharmacogenomic analysis carried out in patient-derived cell lines are often validated in patient derived xenografts (PDXs) and preclinical animal models (a process often called ‘T1-T4 stage’ validations) [81,82]. Once a concordant observation is achieved in many different such models, the therapy is advanced into clinical trials. However, despite this hierarchy of validation process, clinical trials often show discouraging outcomes [83]. Hence, a goal of translational research is to sequentially narrow down as many candidate therapies as possible. In this regard, as our ability to test thousands of compounds and natural products in hundreds of cell lines and animal models has proven useful [15,84].

Preclinical research has also paved the way for predicting pharmacokinetics and pharmacodynamics of drugs. For example, the effective dose of a drug required in the patient can be measured accurately by comparing QSAR and IC50 properties of drugs with other optimized drugs. Moreover, doses of a wide variety of drugs measured in isogenic cell lines settings can successfully recapitulate the dose spectrum required in syngeneic mouse models and even in human patients [15,71,85].

Although, the link between IC50 at the cellular level and in vivo at the plasma concentrations may sometimes be complicated, a general equation linking in vivo doses and effective concentrations is:(D/τ) = (CL/F) × C_target_
where: D = Dose, τ = dosing interval, CL = body clearance of the drug, F = bioavailabilty (fractions absorbed) of the drug by the selected route of administration and C_target_ = total plasma concentration required for desired effect [86]. The C_target_ is usually greater than IC50, since desired response often is more than just 50% of inhibition of the system.

Furthermore, precise in vitro genetic modification of patient derived T-cells has resulted in breakthrough efficacy in the form of immunotherapy [87]. In this regard, the efficacy of immunotherapeutic agents- ‘checkpoint inhibitors’, have also been widely facilitated by mechanistic insight gained by in vitro and preclinical research [88].

## 13. Challenges and Scopes

Quantitative chemical biology research is not without limitations. For example, HTS performed in cell lines has not often faithfully recapitulated the complexity of the microenvironment, cell-tissue heterogeneity and host-microbiome interactions in mouse models or human patients [89,90]. To address this, the field is increasingly experimenting with 3D and co-culture driven models [91,92]. As mentioned before, a caveat of biomarker and candidate target prediction approaches is that it often is correlative and lacks causality. With the advent of artificial intelligence, the use of machine learning is proving useful to predict causal biomarkers more accurately for therapy [23]. However, machine learning is limited in its ability to nullify batch artifacts as well as to combine different OMICS data—challenges to be addressed in the future.

For causality analysis, the recent advent in CRISPR technology has made it possible to perform high throughput loss of function genetic screens and couple that information with chemical perturbagen screens in order to improve target discovery [34,59]. Variations of CRISPR technologies, such CRISPRa, has made it possible to conduct gain of function screens in this context as well [37,38,59].

One of the major limitations of current cancer therapies is the emergence of resistance. Accumulating evidence indicates not only mutation of the drug target but also the presence of co-occurring mutations and heterogenous genetic and epigenetic background of tumors as causative factors promoting drug resistance [93,94]. Acquisition and analysis of -OMICS data serially to characterize tumor evolution during treatment has the potential to provide target-driven therapeutic approaches. One goal would be to predict the trajectory of tumor evolution during treatment based on preclinical and clinical data and deploy and adapt the therapy regimen accordingly [83,95].

## 14. Conclusions

In this review, we have attempted to summarize recent developments in quantitative chemical biology and outlined parameters for such quantitative assays. Successful implementation of cancer therapeutics requires a comprehensive understanding and analysis of both -OMICS data and pharmacodynamic-pharmacokinetic responses [83]. We hope that an improved quantitative understanding of chemical biology will transform aggressive cancers into chronic or curable conditions through more accurate clinical use of current and future systemic therapies.

## Figures and Tables

**Figure 1 cancers-14-05254-f001:**
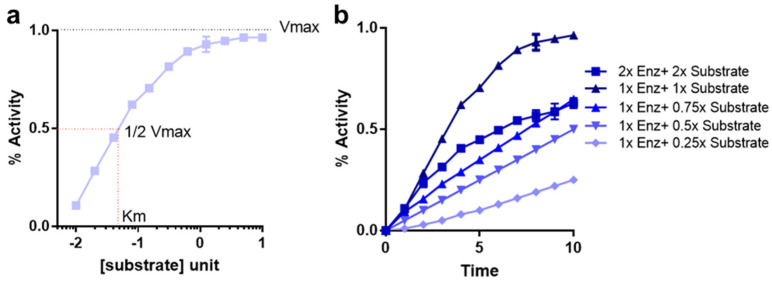
Reaction kinetics and velocity plots. (**a**) Michaelis-Menten reaction rate is plotted as a function of substrate concentration. Reaction rate is saturated at Vmax. (**b**) Reaction progress curve for initial velocity measurement under a varying concentration of substrates and enzyme as indicated.

**Figure 2 cancers-14-05254-f002:**
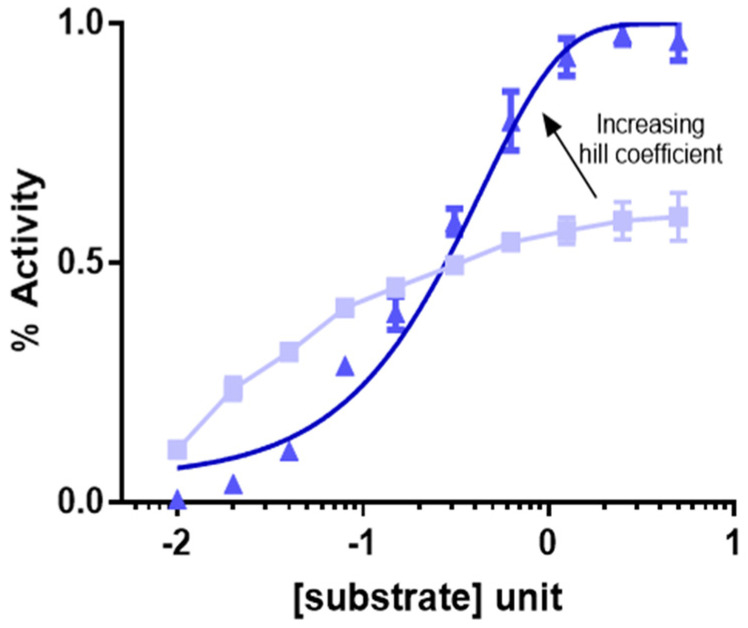
Hill co-efficient and sigmoidal curve. The light blue curve represents a classical Michaelis-Menten reaction kinetics, whereas the darker blue curve represents transformation of the Michaelis-Menten reaction kinetics in presence of cooperativity and hill coefficient. The steeper slope that the inflexion of the sigmoidal curve has the higher the hill coefficient for co-operativity.

**Figure 3 cancers-14-05254-f003:**
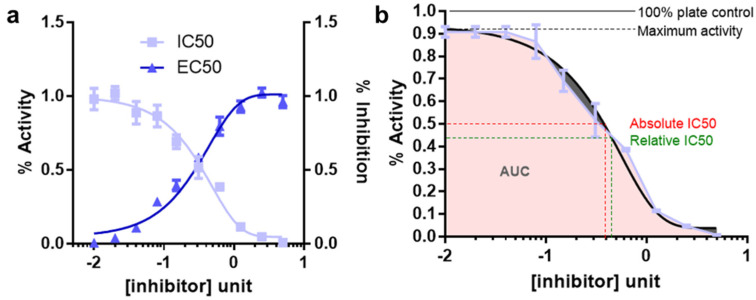
IC50, EC50 and AUC measurements from dose response plots. (**a**) Dose response plots for AC50 (dark blue) and IC50 (light blue). (**b**) AUC calculation of a dose response plot. The figure also represents the theoretical (red) and relative/tested (green) IC50 measurements from the dose response curves. The theoretical (solid black line) activity represented as plate control and maximum activity tested (broken black line) can also be different.

**Figure 4 cancers-14-05254-f004:**
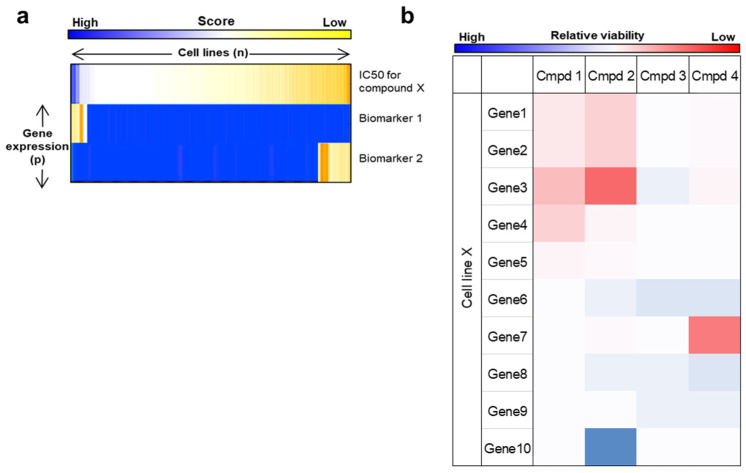
From heatmap to actionable OMICS feature selection. (**a**) Heatmap representation of a typical regularized regression (elastic net) driven dose response versus biomarker plot. IC50 of many different drugs in a panel of cell lines is plotted on the first row and the other row 2 represents anti-correlative biomarkers and row 3 represents correlative biomarkers. Biomarker solutions are rank ordered based on their score. (**b**) Heatmap representation of chemigenomic interaction in an isogenic setting. Many genetic alterations in ‘Cell line X’ is presented on rows whereas columns represent viability of the genetically modified cells under different compounds (columns).

**Figure 5 cancers-14-05254-f005:**
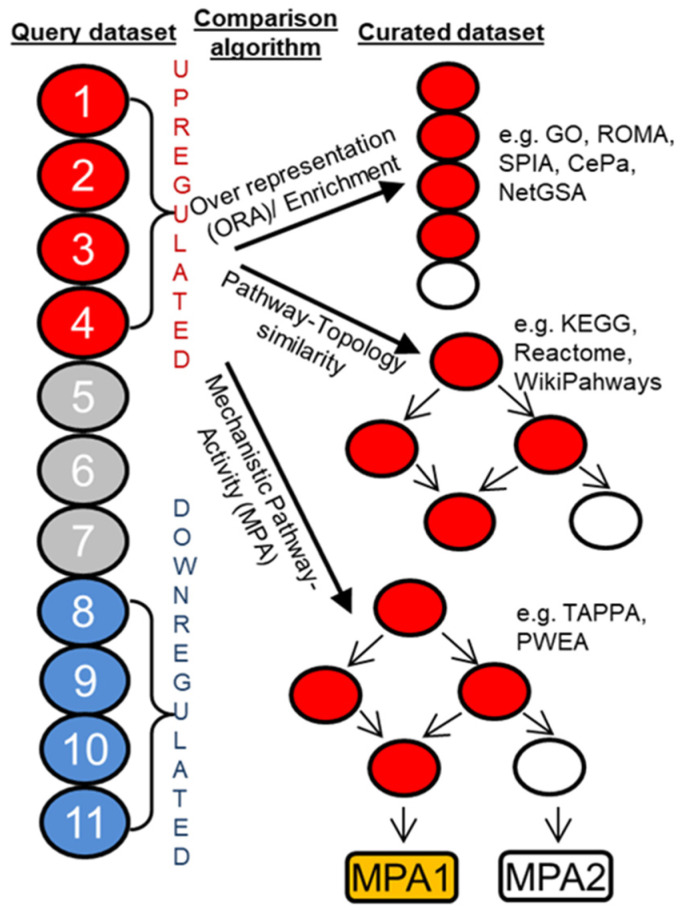
Pathway analysis methods. Comparison of test dataset with curated databases can be performed in many ways- based on over representation, pathway topology-based, mechanistic-pathway activity.

**Figure 6 cancers-14-05254-f006:**
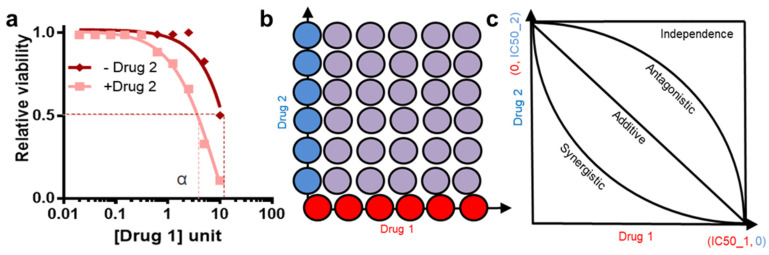
Drug synergy measurement methods. (**a**) Synergy measured by visualizing left-shifting of dose response curve of Drug 1 in presence of the Drug 2. (**b**) Cartoon illustrating comprehensive checkerboard- style experimental setup where combinatorial doses are marked as purple and only Drug 1 as red and only Drug 2 as blue. (**c**) Isobologram analysis indicates multiple possible outcomes for drug interaction analysis- for additivity diagonal, independence rectangular, for synergy concave and for antagonistic convex isobolograms.

**Table 1 cancers-14-05254-t001:** Comparison between phenotypic and target-based screening.

	Phenotypic Screening	Target Based Screening
**Molecular targets**	Not known	Known
**MOA**	Not known, but can be targeted based on signaling pathways	Known
**Assay type**	Cell viability (e.g., luminescence read out live cells)	Direct binding assays (e.g., fluorescence read out in FRET)
**Assay scale**	Relatively difficult to scale up	Easily scalable into high throughput
**Biological relevance**	Highly relevant to biology	May not be relevant to functional biology
**Quantification methods**	Not available	Structure activity relationship (SAR)
**Novel target scope**	High	Low

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
