# Peer review of "Quantitative Framework for Bench-to-Bedside Cancer Research"

_cancers, 2022, doi:10.3390/cancers14215254_

Round 1

Reviewer 1 Report

The manuscript “Quantitative Framework for Bench-to-Bedside Cancer Research and Treatment” by Zaman and Bivona is an excellent review highlighting modern cancer research in vitro methods. The extensive methodology discussed will be helpful for anyone starting in research or already well experienced cancer researchers.

Suggestions:

1.       The title is a bit misleading, suggest changing it into:” Quantitative Framework for Bench-to-Bedside Cancer Research”. There is very little discussion about treatment and the last section (11) of the review would benefit from expansion and some examples.

2.       Since there is significant emphasis on cell viability screening and data it would be worthwhile mentioning methods for assessing it. Only cellTiterglo is mentioned but there are other well used methods with their own caveats.

Author Response

Reviewer 1:

The manuscript “Quantitative Framework for Bench-to-Bedside Cancer Research and Treatment” by Zaman and Bivona is an excellent review highlighting modern cancer research in vitro methods. The extensive methodology discussed will be helpful for anyone starting in research or already well experienced cancer researchers.

Suggestions:

  1. The title is a bit misleading, suggest changing it into:” Quantitative Framework for Bench-to-Bedside Cancer Research”. There is very little discussion about treatment and the last section (11) of the review would benefit from expansion and some examples.

We thank the reviewer for taking interest in the overall work and for the insightful comment. We have now changed the title as per his request. We have also added a case study to discuss a real world scenario where these quantitative techniques can aid therapy discovery and refinement. We believe this improves our manuscript (attached).

  1. Since there is significant emphasis on cell viability screening and data it would be worthwhile mentioning methods for assessing it. Only cellTiterglo is mentioned but there are other well used methods with their own caveats.

We thank the reviewer for the comment. We have now added a paragraph outlining many other viability measuring methods to address this suggestion.

Reviewer 2 Report

This paper presents a series of quantitative techniques to layout a basic framework for bench-to-bedside quantitative research and therapy. It covers Michaelis-Menten (MM) model for drug dose response, IC50 to measure inhibitor effects, high throughput screen (HTS) for pharmacological compounds, machine learning for biomarker prediction, isogenic system for comprehensive chemogenomic analysis, pathway analysis for new discovery, quantitative structure activity relationship to define drug physicochemical properties, determination of drug synergy by IC50 in various models, and bench-to-bedside translation with discussion of challenges. 

While the concept of quantitative framework is innovative, description/illustration is limited. At the end, it is difficult for reader to understand the ‘Quantitative Framework’ and the connection of all the components to this framework. An integrative diagram with real example could be informative.  Similarly, adding real example in each component and its connection to other components could improve the value because description in most parts is technical or generic.    

Author Response

Reviewer 2:

This paper presents a series of quantitative techniques to layout a basic framework for bench-to-bedside quantitative research and therapy. It covers Michaelis-Menten (MM) model for drug dose response, IC50 to measure inhibitor effects, high throughput screen (HTS) for pharmacological compounds, machine learning for biomarker prediction, isogenic system for comprehensive chemogenomic analysis, pathway analysis for new discovery, quantitative structure activity relationship to define drug physicochemical properties, determination of drug synergy by IC50 in various models, and bench-to-bedside translation with discussion of challenges. 

While the concept of quantitative framework is innovative, description/illustration is limited. At the end, it is difficult for reader to understand the ‘Quantitative Framework’ and the connection of all the components to this framework. An integrative diagram with real example could be informative.  Similarly, adding real example in each component and its connection to other components could improve the value because description in most parts is technical or generic.    

We thank the reviewer for the comment. To address the issue of connecting the concepts, discussed in our manuscript, to a real world scenario, we now have added a paragraph with a hypothetical case study with reference to appropriate figures (attached). We believe this improves our manuscript and we thank the reviewer for the important insight.

Round 2

Reviewer 2 Report

The revision includes a case study to address the issue of connecting the concepts. However, authors seem to overlook the problem by providing a generic and less informative case study. 

Author Response

Dear editor,

Thank you for the comments. We have now revised the manuscript to include a more detail and real-life case-study. In the case study we discuss a case where FGFR1 upregulated tumors responded better upon FGFR and MEK inhibitor polytherapy.  In this version of the case study, we also explain the rationale and alternatives for each experimental step in more detail. We believe this has improved our manuscript. Please let us know what you think.

Best.
Aubhishek
